# Electroencephalografic Activity during the Reading of Erotic Texts with and without Aggression

**Claudia Amezcua-Gutiérrez** [1,*], **Marisela Hernández-González** [1], **Enrique Hernández-Arteaga** [2], **Rosa María Hidalgo-Aguirre** [3] **and Miguel Angel Guevara** [1]

1. Institute of Neuroscience, University Center for Biological and Agricultural Sciences (CUCBA), University of Guadalajara, Guadalajara C.P. 44130, Jalisco, Mexico; marisela.hgonzalez@academicos.udg.mx (M.H.-G.); maguevara1954@gmail.com (M.A.G.)
2. Faculty of Human Development, Autonomous University of Tlaxcala, Tlaxcala de Xicohténcatl C.P. 90000, Tlaxcala, Mexico; eharteaga@uatx.mx
3. Health Sciences Department, Valleys University Center (CUVALLES), University of Guadalajara, Ameca C.P. 46708, Jalisco, Mexico; rosa.hidalgo@academicos.udg.mx
* Correspondence: delcarmen.amezcua@academicos.udg.mx

**Abstract:** Sexual arousal (SA) is a multidimensional experience that includes cognitive, emotional, motivational and physiological components. Texts with erotic content have been used to generate a state of SA. Erotic texts often include aggressive content that has not been evaluated in relation to SA. The aim of this work was to compare cortical functionality in women when reading a sexually explicit text (SET) and a sexually explicit text with aggression (SETA). Twenty-seven women participated. The EEG activity of the frontal, temporal and parietal locations was recorded during the reading of both texts. The participants found the SET to be more pleasant than the SETA. Both texts were identified as triggers of general and SA. While reading the SETA, there was an increase in absolute power in the frontal and parietal locations, a higher intrahemispheric correlation between the left frontal and temporal locations in fast frequency bands and a greater interhemispheric correlation between the frontal locations in the delta and alpha1 bands. These findings indicate that cortical functionality during SA in women differs based on the content and context of the erotic material being read, possibly associated with mechanisms that underlie the processing and incentive value assignment of stimuli with sexual and aggressive connotations.

**Keywords:** sexual arousal; EEG; erotic reading; erotic–aggressive reading; women





## 1. Introduction

In several experimental works, different kinds of sensory stimuli (tapes, videos or photos and sexually explicit texts) have been used to generate sexual arousal in humans [1–3]. Particularly, a sexually explicit text (SET) describes in detail a sexual experience between a young couple. It includes flirting, petting and foreplay including oral–genital stimulation, coitus in various positions, orgasms and post-coital behavior [4].

Jakobovits [3] found that women are more aroused by hardcore pornography than by what he terms "erotic realism". The hardcore pornography stories were differentiated from erotic realism in that they contained greater sexual concentration (number of sexual terms), more exaggeration (use of superlatives) and fewer anti-erotic elements (references to situations and emotions incompatible with or detrimental to sexual stimulation).

Other studies have tended to support the findings of Jakobovits. For example, Schmidt et al. [4] asked to 120 female and 120 male students to read and evaluate one of two stories in which the sexual experience of a young couple was described. The stories differed in the degree to which affection was expressed. On average, the stories were rated as "moderately sexually arousing" and the type of story only had a very slight influence on the measured responses. Hence, the authors concluded that affection (romance) was not

a necessary precondition for women to react sexually to sexual stimuli. In another study, Osborn & Pollak [1] asked 12 young female graduate students to read two sets of SET. One set was classified as erotically realistic and the other as hardcore pornography. They measured sexual activation (SA) subjectively using self-report scales and physiologically by means of a vaginal photoplethysmograph. They found that both sets of stories produced sexual arousal. However, the hardcore stories produced significantly greater arousal on the vaginal pressure pulse measure and the subjective report measure. Psychologists have reported that some women find the descriptions of forced sexual submission exciting [5].

Fantasy is a crucial element of erotic reading. Sexual fantasies significantly contribute to sexual arousal. In men, both self-reported and physical arousal were higher after exposure to visual sexual stimuli compared to sexual fantasy stimuli [6]. However, this difference has not been observed in women [7].

Particularly, current erotic texts imply an aggressive component, too. The notion that SA and aggression are closely linked has been studied since the mid-1960s, from three points of view: (1) the perspective that aggression may serve to increase sexual arousal and pleasure [8], (2) the perspective that heightened sexual arousal is highly effective in inhibiting subsequent aggression [9] and (3) the perspective that sexual arousal is likely to enhance behavioral aggression in both men and women [10–12]. As is evident, these approaches are not recent. Recent research revolves around the impact of SA on men's sexual aggression proclivity [13,14], psychopathy and sexual aggression [15] and sexual coercion or forcefulness [16,17].

In summary, the relationship between sexuality and aggression has been empirically studied; however, these approaches evaluate the influence of aggression on SA or vice versa and leave aside the aggressive context embedded in the SA, analyzing both components separately.

Several studies have evaluated the brain response during SA through different imaging techniques such as functional magnetic resonance imaging (fMRI) and positron emission tomography (PET). These studies have demonstrated that several brain areas are activated during SA and orgasm in men and women. These include the amygdala, ventral striatum, caudate nucleus and insula, as well as the prefrontal, parietal and temporal cortical areas [2,18,19].

The PFC has been divided into three areas of study: dorsolateral (DLPFC), orbital (OFC) and medial (mPFC) [20], each of which has been implicated in different aspects of SA.

DLPF participates in motivational/emotional processes related to inhibition as a mechanism that requires the suppression of internal and external information inputs that can influence behavior [21] and the motivational valence or incentive value of stimuli with sexual content [22,23]. OFC is associated with emotional processing and the motivational value of stimuli with sexual content [23], and mPFC has been implicated in the maintenance of representations of stimuli of sexual content in working memory, the integration of sensory information [19,24] and the social properties of the present stimuli and the emotion they evoke [25,26]. For their part, parietal areas have been related with the perception of bodily sensations, such as penile tumescence [2,27,28], and with attention processes directed to relevant motivational stimuli [28,29]. Finally, the temporal lobe, where the amygdala is located [30], plays an important role in regions that are most often involved in the regulation of human sexual behavior [31]; they are also involved in shaping the emotional meaning of sexual stimuli [32], regulating related cognitive and emotional processes.

The perception and processing of emotionally charged stimuli, including sexual stimuli, necessitate the activation and functional coordination of various cortical and subcortical brain structures [2,33]. In this sense, more detailed information on the neural activity and connectivity within these areas is required.

Imaging techniques offer a spatially accurate view and pinpointing of cortical regions associated with sexual behavior; however, electrophysiological methods (EEG) record

neuroelectric activities with superior temporal precision, enabling the formulation of hypotheses regarding neuronal activity and its exact timing down to the millisecond.

Electroencephalographic (EEG) recording is a non-invasive method with a high temporal resolution, enabling the collection of data on brain function across different behavioral, physiological and psychological states.

It consists of a mixture of rhythmic sinusoidal-like fluctuations in voltage generated by the brain and recorded from the scalp surface after being picked up by metal electrodes and conductive media [34]. The frequency spectrum of the EEG in humans is divided into five bands: delta ($\delta$, 1.5–3.5 Hz), theta ($\theta$, 3.5–7.5 Hz), alpha1 ($\alpha1$, 7.5–10.5 Hz), alpha2 ($\alpha2$, 10.5–13.5 Hz), beta1 ($\beta1$, 13.5–19.5 Hz), beta2 ($\beta2$, 19.5–30 Hz) and gamma ($\gamma$, 31–50 Hz).

Low-frequency brain waves, such as delta waves, are typically recorded during the deeper stages of sleep and in states involving motivational urges triggered by biological rewards, attention and the detection of salient stimuli [35]. Theta waves are usually observed during drowsiness and lighter sleep stages, as well as in pleasant [36], relaxed [37] and positive emotional states [38]. Alpha waves are generally recorded in awake individuals with their eyes closed. A reduction in alpha waves indicates increased brain activation linked to processing emotional responses to relevant stimuli [39]. Changes in the alpha range are also associated with attentional processes and the inhibition of irrelevant information [40]. Beta waves appear during arousal [41], during heightened alertness [42] or in response to optimal sensory stimuli. Finally, the gamma band is related to cognitive processing and perceptual experiences [43].

One of the EEG analysis parameters is the absolute power (AP) of each band. AP is defined as the power density of each frequency band expressed in microvolts squared ($mV^2$/Hz). Another analysis parameter is the EEG correlation, which refers to the synchronization among electrical activities at different cortical fields [44], which has been used in several studies of humans to determine whether the electroencephalographic connectivity between brain regions changes in relation to specific states [45].

To date, many works on sexual arousal in humans use different brain-imaging methods and behavioral paradigms. Ziogas et al. [46] carried out a meta-analysis on neuroelectric measurements or stimulations associated with various forms of human sexual behavior. The primary aim of this work was to deliver an in-depth overview of the topic, identify potential limitations in the field and suggest methodological and theoretical directions for future research. In studies that use the EEG technique, the authors have noted the wide range of methodologies employed. For instance, different types of stimuli are used to generate SA. Researchers emphasize the importance of evaluating the emotional properties of these stimuli and ensuring they vary in emotional valence and arousal based on the participants. Additionally, the studies differ in terms of the modalities (auditory, visual, olfactory or combinations) and the duration of the stimulus presentation [46].

Even though several investigations have addressed this topic, few studies have examined the cerebral basis of SA, particularly in women [1,5], and even fewer have used erotic reading with simultaneous EEG recording [47].

SET induces a state of SA [1,3,4]. Nevertheless, erotic texts often include aggressive content that has not been evaluated in relation to SA. Thus, considering that the processing and incentive value assignment of stimuli with sexual and aggressive connotations depends on the functionality of the prefrontal, parietal and temporal cortices, the aim of the present study was to compare cortical functionality by recording electroencephalographic activity (EEG) in young women while they read a sexually explicit text (SET) and a sexually explicit text with aggression (SETA).

We hypothesize that sexually explicit texts with aggressive content (SETA) will result in different patterns of cortical activation than sexually explicit non-aggressive texts (SET) in young women.

Identifying the degree of activation and synchronization between the prefrontal and posterior cortices during the reading of a sexually explicit text and a sexually explicit text with aggression could help us to understanding the cerebral mechanisms that underlie SA

and to know the functional cortical changes that women require to recognize and adapt their response to this type of sexual stimulation.

## 2. Materials and Methods

### 2.1. Subjects

To select the participants, semi-random probabilistic sampling was carried out. The sample consisted of 27 healthy heterosexual women aged 20–30 (mean, 24.03; SE = 0.59), all of whom were right-handed, had a minimum of 13 years of schooling, had an average reading ability and reported no previous brain disease, psychopathology, neural injury or presence of sexual dysfunction. None were under any type of medication or drug known to influence EEG recording. They were asked to refrain from drinking caffeine or alcohol during the 12 h prior to the recording sessions. The participants were recruited by personal invitation or over the Internet by e-mail or social media and invited to participate in a study voluntarily and without economic compensation, in which their brain electrical activity would be recorded during the reading of a sexually explicit text. All participants were assured that confidentiality would be maintained and that they were free to withdraw from the experiment at any time without penalty.

All recordings were made between post-menstrual days 4 and 8. The study was approved by the Institute's Committee on Research Ethics and complied with all APA ethical standards. The subjects were only allowed to participate after giving their informed consent.

### 2.2. Stimuli

Two types of texts were utilized: one categorized as sexually explicit (SET) and the other as sexually explicit with aggression (SETA). The selection was based on a previous pilot study involving 14 young heterosexual women (average age = 23.6 years). The SET comprised excerpts from the Fifty Shades trilogy [48–50]. Given that films depicting heterosexual vaginal intercourse are generally found most appealing and arousing by women [51], the excerpts included descriptions of sexual activity performed by a young heterosexual couple, featuring anatomical references, passionate kissing, fondling and explicit sexual descriptions. In the case of SETA, it comprised excerpts from the aforementioned trilogy, which added whipping, spanking and bondage.

Both sets of texts were matched in terms of writing style and were connected to ensure coherence. Each text consisted of approximately 700 words, divided into seven paragraphs of 70–100 words each. These paragraphs were displayed sequentially in white text on a black computer screen with a resolution of 480 × 640 pixels. The font used was Times New Roman, size 22. The participants were given 30 s to read each paragraph, resulting in a total reading time of 3 min per text.

### 2.3. Questionnaires and Scales Utilized

To ensure that the women studied were heterosexual, they answered the Kinsey's Scale [52] at the beginning of the experiment. The Kinsey scale, also known as the Heterosexual–Homosexual Rating Scale, is utilized in research to describe an individual's sexual orientation based on their experiences or responses at a specific moment. This scale ranges from 0 (for those who identified themselves as exclusively heterosexual) to 6 (for those who self-identify as exclusively homosexual). Ranges of 1–5 indicate women who recognize varying levels of desire for sexual activity with either sex, including "incidental" or "occasional" desires for sexual activity with the same sex. This study included only heterosexual women who reported scores of 0 or 1.

To obtain a measure of the valence and degree of both general and sexual arousal while reading the texts, EEG recording was performed during the reading, and the subjects answered two scales immediately after finishing each text. The first was the Manikin Self-Assessment Scale (SAM) [53], a non-verbal, pictorial questionnaire designed to directly assess a person's emotions and feelings in response to an object or event, such as a picture. This technique is extensively used by scientists in psychology experiments to gauge partici-

pants' emotional reactions due to its non-verbal format. The questionnaire features three rows of pictograms, each depicting a five-point scale across three domains: valence, arousal and dominance. For this work, only valence and general activation were evaluated.

Valence (sometimes referred to as "pleasure") involves concepts like Unhappy–Happy, Annoyed–Pleased and Unsatisfied–Satisfied. Images ranging from a smiling, happy figure to a frowning, unhappy one represent the pleasure dimension (stimuli rated 1–3 were considered "unpleasant"; 4–6, "neutral"; and 7–9, "pleasant"). General arousal is related to concepts such Relaxed–Stimulated, Calm–Excited and Sluggish–Frenzied and was rated using images that showed an excited, wide-eyed figure and a relaxed, sleepy one. Here, ratings of 1–5 were considered "not activated", and ratings of 6–9 were considered "activated". The test–retest reliability coefficient for the SAM and researcher-made test was in the range of 0.55–0.78. The concurrent validity ranged from 0.56 to 0.87, and the criterion validity was acceptable [54].

The next scale was the Sexual Arousal Scale (SAS), which is based on the principles of the Manikin Self-Assessment Scale (Likert-type scale). It was used to indicate the degree of subjective sexual arousal or sexual activation, which refers to the physiological changes that the body undergoes when we feel desire—in this case, vaginal lubrication after reading the SET and SETA. It consists of a series of five drawings of smiling faces of different intensities that represent vaginal lubrication, where 1 = no lubrication (no sexual activation) and 9 = very high lubrication (high sexual activation). This allowed us to obtain a quantitative measure of the subjective experience of sexual arousal. The utility of this scale has been demonstrated in previous studies [55].

Women's sexual functioning was assessed using the "Female Sexual Function Index" (FSFI), a 19-item questionnaire designed as a concise, multidimensional self-report tool that evaluates six dimensions of sexual function in women: desire, arousal, lubrication, orgasm, satisfaction, and pain, along with a total score. The scores reflect the sexual experience over the past four weeks on a scale from 1 to 6, where 1 represents the highest value and 6 represents the lowest or null. A score of 26.55 or below indicates female sexual dysfunction. The internal reliability for the total FSFI and six domain scores is good to excellent, with a Cronbach alpha's > 0.9 for the combined sample and one above 0.8 for the sexually dysfunctional and non-dysfunctional samples, independently [56,57].

### 2.4. EEG Recordings and Procedure

All recordings were made in a sound-attenuated room at 22–23 °C, between 10:00 and 15:00 h. Electrodes were placed on the right and left prefrontal (F3, F4), temporal (T3, T4) and parietal (P3, P4) locations according to the 10–20 international system [58]. The reference used was linked ears with impedances kept below 10 kΩ to minimize the reference electrode and volume conduction effects on EEG correlations. EEG data were recorded using a Grass P-7 polygraph (Grass Technologies, Co., West Warwick, RI, USA), with filters set between 1 and 60 Hz, digitized at a sampling rate of 512 Hz and saved on a PC via CAPTUSEN acquisition software [59]. The participants were seated comfortably in front of a laptop in a closed room. EEG recordings were conducted under two conditions: reading the SET and SETA texts, each for a duration of 3 min.

EEG signals were analyzed off-line with CHECASEN software [60], which displays the EEG epochs on a computer screen. By moving two cursors along the EEG signals, it is possible to select and store artifact-free EEG segments (caused by eye movements, muscle activity or heartbeats) that corresponded to the specific periods of interest. Thus, 60 epochs were selected from each text for each participant. Fast Fourier Transform (FFTs) analyses were applied to the artifact-free data in 1 s epoch samples, with the spectral graph ranging from 1 to 60 Hz at a 1 Hz resolution. The FFTs were performed using EEGmagic software [61]. Absolute power (AP), defined as the power density of each frequency band expressed in microvolts squared ($mV^2/Hz$), was calculated for each condition and the traditional EEG bands: delta, δ (1–3 Hz), theta, θ (4–7 Hz), alpha1, α1 (8–10 Hz), alpha2, α2 (11–13 Hz), beta1, β1 (14–19 Hz), beta2, β2 (20–30 Hz) and gamma, γ (31–50 Hz).

EEG synchronization or correlation is a mathematical index that makes it possible to calculate the degree of similarity between two EEG signals in different frequency bands or at specific frequencies. It is sensitive to both phase and polarity, regardless of amplitude [44]. Thus, the correlation between cortices between hemispheres (interhemispheric correlation or rTER) and in the same hemisphere (intrahemispheric correlation rINTRA) was also calculated for each one of the six frequency bands by means of Pearson Product-moment correlation analysis.

### 2.5. Experimental Design and Statistical Analyses

The experimental design was a Correlated Group Design (within-subjects or repeated measures) given that all participants were exposed to all the conditions.

The parameters of valence, general arousal and sexual arousal between conditions (SET and SETA) were compared by Wilcoxon tests.

To determine the EEG differences between conditions, a student's *t* test was performed for groups correlated with the PA, rTRA and rTER values of the frontal, temporal and parietal locations. Values with $p \leq 0.05$ were considered significant (for each of the traditional EEG bands between the different conditions).

For statistical purposes, the AP values were transformed into logarithms [62], while the correlation values were transformed to Fisher z scores.

Similarly, the Spearman correlation values between the absolute power values of each of the bands (delta, theta, alpha 1, alpha 2, beta 1, beta 2 and gamma) and the reported values from the Manikin self-assessment scale (valence, general activation) and Sexual Arousal Scale were calculated. From these values, a significance level of $p < 0.05$ was established, indicating that the correlation value is statistically significantly different from zero.

### 3. Results

#### 3.1. Scales

As mentioned above, only women who answered 0 or 1 on the Kinsey scale, i.e., exclusively heterosexual, were included. None of the participants presented sexual dysfunction problems according to the scores of the "Female Sexual Function Index" (FSFI). The mean score was 31.144 ($\pm$1.28), which corresponds to optimal sexual functioning.

With respect to the evaluation of the texts, the results of the Manikin Self-Assessment Scale indicated significant between-condition differences. For valence, the SET was rated as more enjoyable compared to the SETA (W = −13.00; p(W) = 0.0010). Both texts were classified as generators of general activation and sexual activation (Figure 1), and no differences between conditions were obtained (W = −60; p(W) = 0.2628, W = −69; p(W) = 0.0619, respectively).

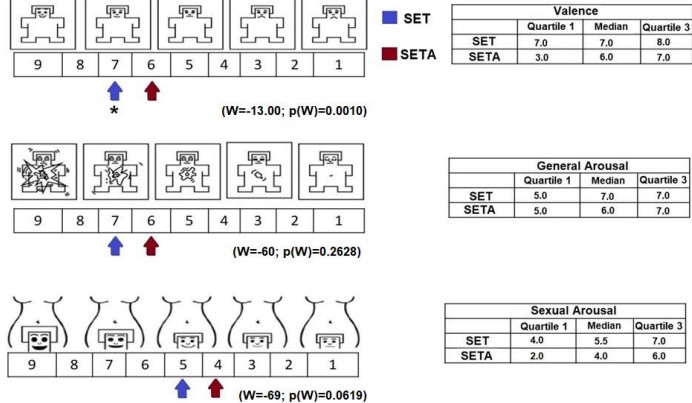

**Figure 1.** On the left side, representation of valence, general activation and sexual activation on the SAM and SAS scales reported by the participants after reading the SETA and the SET. The values are represented with arrows with different colors for each condition. The table on the right side shows the median values and interquartile ranges reported after reading both texts. * p(w) $\leq$ 0.05.

*3.2. EEG*

Statistical analyses revealed significant differences between conditions in the different EEG parameters. In the following section, they will be described, comparing each of the conditions (SETA vs. SET).

3.2.1. Absolute Power

During the reading of the SETA, a higher AP in the alpha1, alpha2 and beta1 bands in the frontal (F3 and F4) and parietal (P3 and P4) locations was obtained. In addition, also during the reading of the SETA, a higher AP of the theta band in P3 and of the gamma band in P4 regarding the SET was observed (Figure 2). Only significant t-values and *p*-values, of each EEG band and the derivations, are mentioned in Table 1.

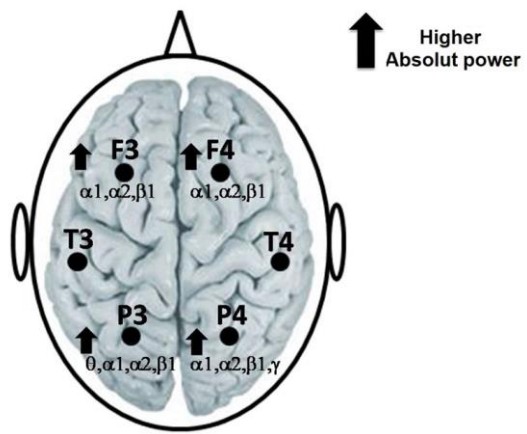

**Figure 2.** Representation of the highest AP of the alpha1, alpha2 and beta1 bands in F3 and F4 of theta, alpha1, alpha2 and beta1 in P3, as well as of alpha1, alpha2, beta1 and gamma in P4, during the reading of the SETA regarding the SET.

**Table 1.** Significant t student values for paired groups, between the different conditions, for the AP values of the frontal, temporal and parietal locations. Values with $p \leq 0.05$ were considered significant for each of the traditional EEG bands (delta, theta, alpha 1, alpha 2, beta 1, beta 2 and gamma).

| Brain Structure Area (Derivation) | Delta | Theta | Alpha 1 | Alpha 2 | Beta 1 | Beta 2 | Gamma |
|---|---|---|---|---|---|---|---|
| Left frontal (F3) | | | t = −3.770 $p = 0.00085$ | t = −3.059 $p = 0.00510$ | t = −2.705 $p = 0.01188$ | | |
| Right frontal (F4) | | | t = −4.221 $p = 0.00026$ | t = −3.108 $p = 0.00452$ | t = −2.429 $p = 0.02234$ | | |
| Left parietal (P3) | | t = −2.603 $p = 0.01415$ | t = −3.359 $p = 0.00311$ | t = −3.550 $p = 0.00149$ | t = −3.889 $p = 0.00062$ | | |
| Right parietal (P4) | | | t = −4.026 $p = 0.00044$ | t = −4.705 $p = 0.00007$ | t = −3.352 $p = 0.00159$ | | t = −2.450 $p = 0.02131$ |

3.2.2. Interhemispheric Correlation

Compared to the reading of SETA, reading SET generated a greater degree of synchronization (rINTER) between prefrontal cortices. This higher correlation was found in delta (t = −2589 $p = 0.015$) and apha1 (t = −3417 $p = 0.002$) (Figure 3A).

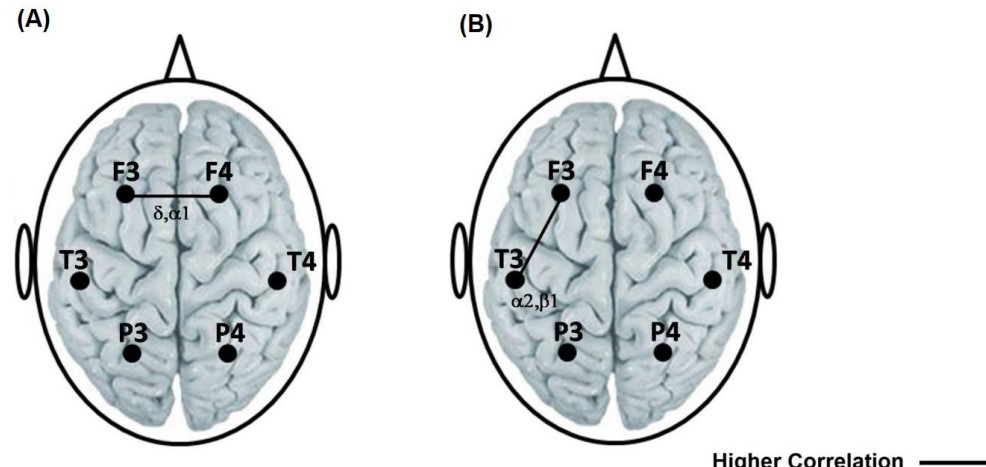

**Figure 3.** (**A**) Representation with a solid line of the highest interhemispheric correlation between the frontal locations in the delta and alpha1 bands during the reading of the TSEA with respect to the TSE. (**B**) Representation with a solid line of the highest intrahemispheric correlation between the frontal–temporal left in alpha2 and beta 1 during the reading of the TSEA with respect to the TSE. The left side is a representation of valence, general activation and sexual activation on the SAM and SAS scales reported by the participants after reading the SETA and the SET.

### 3.2.3. Intrahemispheric Correlation

Compared to the reading of the SET, reading the SETA generated a higher left fronto-temporal correlation in alpha2 (t = −3091 $p$ = 0.004) and beta 1 (t = −3542 $p$ = 0.001) (Figure 3B).

### 3.3. Spearman Correlation among EEG Activity and Self-Assessment Responses

Statistical analyses revealed significant associations between EEG activity and self-perception responses in the different conditions (SETA vs. SET).

### 3.3.1. Sexually Explicit Text (SET)

In general, an inversely proportional relationship was found between the absolute power values of the delta band in both hemispheres of the prefrontal cortex and the responses on the Manikin scale. Similarly, an inverse relationship was observed between the absolute power of the delta band in the left parietal cortex and general activation, as well as between the absolute power of the beta 2 band in this same cortex and sexual activation. In the right parietal cortex, there was an inverse association between the absolute power of the delta band and general and sexual arousal (Table 2). No significant correlation values were found in the other comparisons.

**Table 2.** Spearman correlation (r) values among the AP of the different EEG bands and the data of the Manikin self-assessment scale for the sexually explicit text.

| Brain Structure | Manikin Self–Assesment Scale | Delta | Theta | Alfa 1 | Alfa 2 | Beta 1 | Beta 2 | Gamma |
|---|---|---|---|---|---|---|---|---|
| | Valence | **r = −0.414** **$p$ = 0.0397** | r = −0.284 $p$ = 0.1686 | r = −0.256 $p$ = 0.2167 | r = −0.309 $p$ = 0.1330 | r = −0.392 $p$ = 0.0526 | r = −0.263 $p$ = 0.2043 | r = −0.004 $p$ = 0.9849 |
| Left frontal (F3) | General arousal | **r = −0.577** **$p$ = 0.0025** | r = −0.264 $p$ = 0.2023 | r = −0.160 $p$ = 0.4447 | r = −0.163 $p$ = 0.4371 | r = −0.101 $p$ = 0.6305 | r = −0.097 $p$ = 0.6436 | r = 0.077 $p$ = 0.7144 |
| | Sexual arousal | **r = −0.580** **$p$ = 0.0024** | r = −0.339 $p$ = 0.0972 | r = −0.104 $p$ = 0.6223 | r = −0.068 $p$ = 0.7450 | r = −0.073 $p$ = 0.7302 | r = −0.112 $p$ = 0.5947 | r = 0.045 $p$ = 0.8305 |

**Table 2.** *Cont.*

| Brain Structure | Manikin Self–Assesment Scale | Delta | Theta | Alfa 1 | Alfa 2 | Beta 1 | Beta 2 | Gamma |
|---|---|---|---|---|---|---|---|---|
| Right frontal (F4) | Valence | r = −0.289 p = 0.1613 | r = −0.329 p = 0.1082 | r = −0.134 p = 0.5239 | r = −0.232 p = 0.2635 | r = −0.340 p = 0.0966 | r = −0.060 p = 0.7761 | r = −0.036 p = 0.8636 |
| | General arousal | **r = −0.435 p = 0.0297** | r = −0.297 p = 0.1495 | r = −0.025 p = 0.9055 | r = −0.267 p = 0.1966 | r = −0.364 p = 0.0739 | r = −0.225 p = 0.2791 | r = −0.082 p = 0.6961 |
| | Sexual arousal | **r = −0.420 p = 0.0367** | r = −0.354 p = 0.0824 | r = −0.003 p = 0.9903 | r = −0.132 p = 0.5293 | r = −0.245 p = 0.2384 | r = −0.101 p = 0.6316 | r = 0.048 p = 0.8189 |
| Left temporal (T3) | Valence | r = −0.136 p = 0.5166 | r = −0.221 p = 0.2892 | r = −0.177 p = 0.3972 | r = −0.280 p = 0.1758 | r = −0.061 p = 0.7726 | r = −0.014 p = 0.9473 | r = −0.007 p = 0.9744 |
| | General arousal | r = −0.307 p = 0.1360 | r = −0.206 p = 0.3229 | r = 0.047 p = 0.8240 | r = −0.011 p = 0.9602 | r = 0.042 p = 0.8402 | r = 0.111 p = 0.5967 | r = 0.045 p = 0.8299 |
| | Sexual arousal | r = −0.288 p = 0.1626 | r = −0.253 p = 0.2217 | r = 0.002 p = 0.9911 | r = 0.051 p = 0.8101 | r = 0.119 p = 0.5716 | r = 0.240 p = 0.2479 | r = 0.205 p = 0.3263 |
| Right temporal (T4) | Valence | r = 0.018 p = 0.9308 | r = −0.004 p = 0.9834 | r = −0.163 p = 0.4369 | r = −0.115 p = 0.5840 | r = −0.124 p = 0.5543 | r = 0.050 p = 0.8134 | r = −0.061 p = 0.7732 |
| | General arousal | r = −0.238 p = 0.2523 | r = −0.204 p = 0.3283 | r = −0.061 p = 0.7736 | r = −0.107 p = 0.6104 | r = −0.077 p = 0.7142 | r = 0.055 p = 0.7947 | r = −0.031 p = 0.8816 |
| | Sexual arousal | r = −0.117 p = 0.5765 | r = −0.169 p = 0.4195 | r = 0.011 p = 0.9570 | r = 0.029 p = 0.8889 | r = 0.037 p = 0.8596 | r = 0.227 p = 0.2742 | r = 0.114 p = 0.5863 |
| Left parietal (P3) | Valence | r = −0.363 p = 0.0742 | r = −0.330 p = 0.1069 | r = −0.125 p = 0.5517 | r = −0.091 p = 0.6662 | r = −0.200 p = 0.3383 | r = −0.034 p = 0.8717 | r = 0.087 p = 0.6788 |
| | General arousal | **r = −0.405 p = 0.0447** | r = −0.198 p = 0.3437 | r = 0.069 p = 0.7441 | r = −0.130 p = 0.5366 | r = −0.227 p = 0.2751 | r = −0.353 p = 0.0839 | r = −0.230 p = 0.2697 |
| | Sexual arousal | r = −0.290 p = 0.1598 | r = −0.146 p = 0.4853 | r = 0.006 p = 0.9770 | r = −0.191 p = 0.3593 | r = −0.140 p = 0.5055 | **r = −0.404 p = 0.0450** | r = −0.199 p = 0.3407 |
| Right parietal (P4) | Valence | r = −0.236 p = 0.2560 | r = −0.362 p = 0.0752 | r = −0.195 p = 0.3494 | r = −0.147 p = 0.4825 | r = −0.271 p = 0.1899 | r = −0.099 p = 0.6382 | r = 0.011 p = 0.9578 |
| | General arousal | **r = −0.547 p = 0.0046** | r = −0.293 p = 0.1558 | r = −0.026 p = 0.9010 | r = −0.273 p = 0.1873 | r = −0.337 p = 0.0991 | r = −0.273 p = 0.1866 | r = −0.140 p = 0.5057 |
| | Sexual arousal | **r = −0.441 p = 0.0272** | r = −0.238 p = 0.2513 | r = −0.096 p = 0.6491 | r = −0.306 p = 0.1367 | r = −0.251 p = 0.2259 | r = −0.381 p = 0.0601 | r = −0.149 p = 0.4770 |

### 3.3.2. Sexually Explicit with Aggression Text (SETA)

In this case, an inverse association between the absolute power of the alpha 2 band in the left frontal cortex and general arousal was found. On the other hand, a direct relationship between the absolute power of the beta 1 band in the left parietal cortex and sexual arousal, as well as between the absolute power of the right temporal location and sexual arousal, was found. (Table 3). No significant correlation values were found for the other comparisons.

**Table 3.** Spearman correlation (r) values among the AP of the different EEG bands and the data of the Manikin self-assessment scale for the sexually explicit text with agression.

| Brain Structure | Manikin Self–Assesment Scale | Delta | Theta | Alfa 1 | Alfa 2 | Beta 1 | Beta 2 | Gamma |
|---|---|---|---|---|---|---|---|---|
| Left frontal (F3) | Valence | r = −0.120 p = 0.5692 | r = −0.338 p = 0.0981 | r = −0.163 p = 0.4376 | r = −0.245 p = 0.2385 | r = −0.077 p = 0.7138 | r = −0.049 p = 0.8152 | r = −0.039 p = 0.8543 |
| | General arousal | r = −0.149 p = 0.4775 | r = −0.349 p = 0.0876 | r = −0.024 p = 0.9097 | **r = −0.406 p = 0.0442** | r = −0.166 p = 0.4278 | r = −0.115 p = 0.5835 | r = −0.094 p = 0.6561 |
| | Sexual arousal | r = −0.090 p = 0.6687 | r = −0.365 p = 0.0732 | r = −0.091 p = 0.6659 | r = −0.018 p = 0.9334 | r = 0.138 p = 0.5100 | r = 0.107 p = 0.6098 | r = 0.056 p = 0.7890 |

**Table 3.** *Cont.*

| Brain Structure | Manikin Self–Assesment Scale | Delta | Theta | Alfa 1 | Alfa 2 | Beta 1 | Beta 2 | Gamma |
|---|---|---|---|---|---|---|---|---|
| Right frontal (F4) | Valence | r = −0.132 $p$ = 0.5305 | r = −0.224 $p$ = 0.2823 | r = −0.181 $p$ = 0.3870 | r = −0.002 $p$ = 0.9919 | r = 0.173 $p$ = 0.4079 | r = 0.087 $p$ = 0.6801 | r = 0.116 $p$ = 0.5819 |
| | General arousal | r = −0.106 $p$ = 0.6137 | r = −0.188 $p$ = 0.3673 | r = −0.186 $p$ = 0.3738 | r = −0.329 $p$ = 0.1086 | r = −0.138 $p$ = 0.5102 | r = −0.282 $p$ = 0.1721 | r = −0.183 $p$ = 0.3822 |
| | Sexual arousal | r = −0.250 $p$ = 0.2277 | r = −0.339 $p$ = 0.0971 | r = −0.165 $p$ = 0.4309 | r = 0.125 $p$ = 0.5518 | **r = 0.439** **$p$ = 0.0282** | r = 0.310 $p$ = 0.1319 | r = 0.338 $p$ = 0.0981 |
| Left temporal (T3) | Valence | r = −0.111 $p$ = 0.5964 | r = 0.007 $p$ = 0.9749 | r = −0.079 $p$ = 0.7068 | r = −0.275 $p$ = 0.1836 | r = −0.256 $p$ = 0.2169 | r = −0.104 $p$ = 0.6210 | r = −0.149 $p$ = 0.4776 |
| | General arousal | r = −0.188 $p$ = 0.3685 | r = −0.228 $p$ = 0.2737 | r = −0.011 $p$ = 0.9599 | r = −0.100 $p$ = 0.6344 | r = 0.143 $p$ = 0.4952 | r = 0.094 $p$ = 0.6561 | r = 0.00 $p$ > 0.9999 |
| | Sexual arousal | r = 0.036 $p$ = 0.8649 | r = −0.111 $p$ = 0.5968 | r = 0.031 $p$ = 0.8840 | r = 0.241 $p$ = 0.2457 | r = 0.256 $p$ = 0.2160 | r = 0.344 $p$ = 0.0925 | r = 0.301 $p$ = 0.1437 |
| Right temporal (T4) | Valence | r = 0.130 $p$ = 0.5365 | r = −0.031 $p$ = 0.8821 | r = 0.068 $p$ = 0.7474 | r = −0.013 $p$ = 0.9519 | r = 0.099 $p$ = 0.6368 | r = 0.088 $p$ = 0.6774 | r = 0.080 $p$ = 0.7048 |
| | General arousal | r = −0.056 $p$ = 0.7888 | r = −0.019 $p$ = 0.9267 | r = 0.016 $p$ = 0.9407 | r = −0.196 $p$ = 0.3475 | r = 0.109 $p$ = 0.6028 | r = −0.012 $p$ = 0.9555 | r = −0.080 $p$ = 0.7026 |
| | Sexual arousal | r = −0.095 $p$ = 0.6511 | r = −0.206 $p$ = 0.3243 | r = 0.095 $p$ = 0.6523 | r = 0.231 $p$ = 0.2670 | r = 0.333 $p$ = 0.1038 | **r = 0.445** **$p$ = 0.0258** | r = 0.382 $p$ = 0.0597 |
| Left parietal (P3) | Valence | r = 0.187 $p$ = 0.3703 | r = −0.086 $p$ = 0.6839 | r = −0.127 $p$ = 0.5452 | r = 0.055 $p$ = 0.7936 | r = 0.158 $p$ = 0.4503 | r = −0.033 $p$ = 0.8747 | r = −0.096 $p$ = 0.6489 |
| | General arousal | r = −0.109 $p$ = 0.6042 | r = −0.307 $p$ = 0.1354 | r = −0.024 $p$ = 0.9075 | r = −0.245 $p$ = 0.2373 | r = −0.003 $p$ = 0.9874 | r = 0.018 $p$ = 0.9326 | r = 0.248 $p$ = 0.2314 |
| | Sexual arousal | r = −0.214 $p$ = 0.3050 | r = −0.180 $p$ = 0.3897 | r = −0.118 $p$ = 0.5748 | r = −0.059 $p$ = 0.7790 | r = 0.326 $p$ = 0.1120 | r = 0.302 $p$ = 0.1424 | r = 0.286 $p$ = 0.1658 |
| Right parietal (P4) | Valence | r = 0.372 $p$ = 0.0673 | r = 0.055 $p$ = 0.7950 | r = −0.162 $p$ = 0.4390 | r = 0.080 $p$ = 0.7027 | r = 0.168 $p$ = 0.4229 | r = −0.031 $p$ = 0.8835 | r = −0.052 $p$ = 0.8050 |
| | General arousal | r = −0.056 $p$ = 0.7916 | r = −0.166 $p$ = 0.4277 | r = −0.053 $p$ = 0.8002 | r = −0.203 $p$ = 0.3314 | r = −0.016 $p$ = 0.9407 | r = −0.046 $p$ = 0.8254 | r = 0.324 $p$ = 0.1136 |
| | Sexual arousal | r = −0.082 $p$ = 0.6966 | r = −0.069 $p$ = 0.7448 | r = −0.111 $p$ = 0.5972 | r = 0.051 $p$ = 0.8076 | r = 0.376 $p$ = 0.0637 | r = 0.282 $p$ = 0.1713 | r = 0.333 $p$ = 0.1039 |

## 4. Discussion

Human sexuality has been a topic of great interest and curiosity on both anecdotal and empirical levels. Among the various aspects on which empirical research has focused, the electrophysiological correlates of SA stand out. There are many works in this regard (for a review, see Ziogas, et al., 2023) [46]. Neuroelectric signals have been utilized to explore the mapping of genital sensations on the brain, as well as to study fundamental genital functions during sexual arousal and the recording of cortical electrical signals during orgasms. Additionally, research has investigated how cognition interacts with sexual arousal in both healthy individuals and those with atypical sexual preferences. However, it should be noted that the studies in question show great heterogeneity or differences in the experimental design, the characteristics of the sample and the type of stimuli used, among other points. For example, based on the meta-analysis carried out by Ziogas et al. [46], they describe 127 studies about the interaction between cognition and SA in healthy participants, of which 23.62% are only in men, 6.29% are only in women and the rest combine women and men, which does not allow us to clearly elucidate the brain bases of cognitive and emotional functions in each sex. On the other hand, in almost all studies, the stimuli used to generate SA are images, videos or films, none of which present erotic literature. Van't Hof and Cera (2021) suggest that sexual orientation and preferences might influence brain responses to sexual stimuli, particularly in women [7]. This leads us to question whether the stimuli used are optimal in generating a similar SA state in each sex. From this

perspective, the lack of studies that investigate the brain bases of cognitive and emotional functions in women using a stimulus appropriate to them is evident.

There are some empirical reports indicating that women prefer fantasy to erotic videos, and whoever tends to read erotic novels are heterosexual women [63]. This is perhaps because, unlike men, women often find more 'humanity' in stories or novels. The characters have names, stories and their own desires. Literature provides guidelines that highlight the important relationships developing between participants in a sexual relationship. Unlike erotic videos, reading allows you to add or remove elements based on your preferences, creating a personalized fantasy.

Given the importance of sexual behavior for reproduction, it is often assumed that sexual signals are positively received. However, human sexual behavior is more complex. Throughout our history, various experiences, circumstances, and cultural influences affect how we respond to specific sexual stimuli, and these responses are not always positive. For example, Borg et al. (2014) found brain activity indicating an apparent negative reaction to pornography among women. According to the authors, this could be due to societal moral attitudes toward pornography, a lack of positive associations with it or simply the difficulty of integrating new learning with previous experiences [64].

From this point of view, our findings raise questions about whether the changes in EEG results that we found when presented with sexually content stimuli can be assumed to be associated with the assignment of a positive sexual incentive value or are more likely to be related to the processes involved in evaluating the incentive meaning of a possible sexual relationship such as memory, experiences, contexts and other elements. Therefore, it would be interesting in future studies to try to analyze different routes to find ambivalent, positive and negative responses in relation to stimuli with sexual content.

To our knowledge, this is the first study to compare cortical functionality by recording electroencephalographic activity (EEG) in young women while they read a sexually explicit text (SET) and a sexually explicit text with aggression (SETA).

According to the participants' self-reports on the Manikin and sexual arousal scales, the SET was rated as more enjoyable compared to the SETA. However, it is important to highlight that the valence score assigned to the SETA (6) is not unpleasant, since according to the values proposed by the Manikin scale, this score corresponds to an evaluation of the stimulus as "neutral". It seems that even when the text has an aggressive component, adding a sexual component will minimize the aggressive one, which makes the observed text not considered as unpleasant.

Both texts were classified for the women as generators of general activation and sexual arousal. There is no doubt that many studies [1,3,4] confirm the effectiveness of SETs in generating a state of sexual arousal in women; however, these works present erotic videos without aggression. Possibly, the fact that the SETA has also generated sexual arousal is associated with the proposal of some psychologists about some women finding the descriptions of forced sexual submission exciting [5]. In this regard, Moran et al. [65] report that aggressive behavior is thought to be divided into two motivational elements, the first being a self-defensively motivated aggression against threats and the second being hedonically motivated "appetitive" aggression. Appetitive aggression is conceptualized as fundamentally hedonic in character and related to reinforcing qualities of the violent act itself. In this sense, this appetitive element could be regulating or determining that the SETA was not evaluated as unpleasant or aversive.

During the reading of the SETA, regarding the SET, the women showed a higher AP in the alpha1, alpha2 and beta1 bands in frontal locations bilaterally (both F3 and F4) as well as a greater degree of synchronization (rTER) between these prefrontal locations in the delta and alpha1 bands. F3 and F4 make up the so-called dorsolateral region of the prefrontal cortex [66]. This area is related with the integration and evaluation of neural information goal-directed behavior and "cold" executive functions, such as planning, working memory, information maintained in memory (for instance, verbal, spatial or object information) [67], internal spatial representation [68] and inhibitory control [69]. Likewise, these prefrontal

locations participate in the syntactic processing of reading, i.e., the correct combination and order of words in a discourse [70]. All these processes are required to understand narrative texts, i.e., those that combine statements to form a coherent story [70]. The stimuli used in this study were of a narrative nature, so understanding and processing the information they contain require using similar cognitive resources, which are modulated primarily by the prefrontal cortex. Hence, the greater absolute power and higher degree of coupling between the prefrontal locations in the alpha band presented by women when reading the SETA could be associated with the processing and understanding of the narrative stimulus, including the inhibition of the aggressive component so that the prefrontal cortices work together to inhibit the unpleasant aspects of the SETA and hence reach a sexual and arousal activation state. This suggestion could be supported by the fact that both alpha activity (related to attention) [71] and beta activity (associated with positive emotional states induced by reading) [72] showed a higher AP in these prefrontal locations.

In the same way as in frontal locations, in this study, higher APs of the alpha1, alpha2 and beta1 bands in the parietal locations bilaterally were also obtained during the reading of the SETA text. The parietal lobe has been associated with the integration of somatosensory and visual information and, likewise, with attention [73]. Alpha-band activity (8–13 Hz) responds to a stimulus and/or task demand [74] as well as to the relevance and/or difficulty of a task. Thus, considering that, during the reading of the SETA, a higher AP of the alpha band in parietal locations was observed, it could be suggested that the reading of this text was more demanding, perhaps because the participants tried to avoid the aggressive parts of the text or to decrease the unpleasant state that could be induced in them.

One particularly interesting result of this work is a higher left frontal-temporal correlation in alpha2 and beta when the women were reading the SETA as compared to the SET. These data agree with several electroencephalographic studies that have demonstrated the different functionality of the left and right hemispheres associated with sexual arousal [75].

F3 and F4 are prefrontal areas specifically constituting the so-called dorsolateral region. In terms of function, this area is related to goal-directed behavior, temporal sequencing, the evaluation and integration of neural information and cold executive functions such as planning and working memory [68]. In contrast to the dorsolateral prefrontal cortex, temporal regions are most often involved in regulating human sexual behavior [76]. The temporal area also participates in modulating the emotional meaning of sexual stimuli [77] and, therefore, regulates interconnected cognitive and emotional processes [2].

Thus, in our study, the higher degree of coupling between the prefrontal and temporal regions may indicate a major modulation of the cold, logical executive systems (dorsolateral prefrontal cortex) on the emotional processes in the temporal cortex. It has been suggested that the functional dissociation between the prefrontal and posterior cortical locations is required to reach an adequate state of sexual arousal [78]. Hence, it is probable that during the SETA condition, a higher prefrontal–temporal correlation could be associated with a major modulation of sexual arousal so that they can suppress any timidity or discomfort that might interfere with the positive emotional arousal associated with reading erotic text with aggressive references.

Interestingly, our study did not find an EEG correlation between the prefrontal and parietal cortices. This agrees with the findings of a previous study conducted on young women reading either a sexually explicit text or one with neutral content [47]. A possible explanation for this fact is the cognitive processes involved in reading the texts presented. Understanding a narrative passage (i.e., a text with statements that form a coherent story) requires higher-order processes (such as attention and working memory [79]), in which the prefrontal [67] and parietal [80] cortices play an important role. Considering that the women require this cognitive process while reading both SETs and SETAs, it is probable that the possible differences in this EEG parameter were not found in this study.

Posner & Petersen [81] described an anatomical circuit between the prefrontal and parietal areas, which has been widely associated with the attentional processes responsible for achieving and maintaining sensitivity to incoming stimuli. Because of their narrative

nature, the reading of both texts used in this study, regardless of their content, required attentional and memory processes and the integration of concepts. Considering that the prefrontal and parietal cortices participate in these processes, it is possible to suggest that they work together, thus explaining why a similar degree of correlation between them was obtained during the reading of both texts.

Our results disagree with the lower prefronto–parietal correlation observed in men and women during sexual arousal induced by observing visual stimuli with sexual content (photos and videos) [18]. Visual stimulation and reading are processed distinctly. A visual stimulus does not require dealing with spelling, punctuation or the visual images characteristic of written language. Moreover, following Anokhin et al. [18], from an evolutionary perspective, the rapid assessment of a visual scene is critical for an organism's adaptation and survival, quite unlike reading, which is a reconstruction of messages represented by graphic symbols [82]. Thus, it is not surprising to find a different functionality and degree of cortical EEG coupling during sexual activation induced by observing visual stimuli compared to sexual activation induced by reading texts with sexual content.

Regarding the phonological, semantic and orthographic processing of reading, it has been described as the highest level of the delta band associated with alterations in word processing [83]. In our study, negative correlation values (inverse relationship) were observed between delta band activity in the prefrontal cortex and self-reports of valence, general arousal and sexual arousal. This individual differences in women could suggest that women who exhibited higher delta activity might have shown an altered processing of sexual information, probably due to minor attention to that stimulus. This decrease in attention may lead to a lower perception of emotionally charged stimuli, such as erotic content, resulting in reduced emotional and sexual arousal. In this context, high levels of delta waves could act as a filter that diminishes women's ability to fully engage with erotic content, reducing their emotional and sexual response during the reading of erotic texts.

On the other hand, in the case of reading sexual texts with aggression, a positive correlation (directly proportional) was observed between the beta band activity in the prefrontal–temporal locations and sexual arousal. In this regard, the reading of sexual texts with aggression likely induces a state of arousal and heightened alertness in female readers.

Such content can be perceived as stimulating, eliciting increased brain activation, particularly in the prefrontal and temporal locations involved in emotional processing and attention. The heightened activity in the beta band in these areas reflects a more intense processing of aggressive erotic stimuli, consistent with beta waves and the role in states of high alertness and responses to sensory stimuli [41,42]. This brain activation correlates with heightened sexual arousal, indicating that women are not only more alert but also more sexually receptive to this kind of material.

In summary. These data show that cortical functionality during a state of sexual activation in women varies depending on the type of content and the context of the erotic reading. The results of this study expand our knowledge about the cortical mechanisms underlying the processing and incentive value assignment of stimuli with sexual and aggressive connotations among young women.

## 5. Limitations and Future Research

This study is limited because we were unable to recruit a larger sample, primarily due to the difficulty in finding women who fulfilled all the inclusion criteria. The sample comprised predominantly middle-class, well-educated women, factors that place some limitations on our ability to generalize the findings. Although the results of other studies have shown a direct correlation between self-reports of SA generated by texts with erotic content and objective measures of vaginal lubrication, it would be interesting to measure, by plethysmography, the amount of blood in the walls of the vagina while reading both texts.

## 6. Therapeutic or Practical Implications

In addition to being considered a moral problem, sexually explicit materials are considered dangerous due to the medical and social risks that their consumption could generate [84]. However, at present, new suggestions regarding the direction of research on sexually explicit materials have been introduced, arguing for the need to understand the production of sexual meanings, displays and performance norms in contemporary sexually explicit material [85–87].

From this perspective, this study may help inform the associations between reading sexually explicit material and cortical functionality involved in the processing of sexual stimulation and its ability to generate an optimal level of sexual activation.

Therapists could encourage couples to use sexual stimulation like a skill to obtain an optimal level of sexual activation—for example, in couples with sexual dysfunctions related to sexual arousal.

**Author Contributions:** Conceptualization, C.A.-G., M.H.-G. and M.A.G.; methodology, C.A.-G., M.A.G. and E.H.-A.; formal analysis, M.A.G., C.A.-G., M.H.-G., E.H.-A. and R.M.H.-A.; data curation, M.A.G. and C.A.-G.; writing—original draft preparation, C.A.-G., M.A.G., M.H.-G., E.H.-A. and R.M.H.-A.; writing—review and editing, C.A.-G., M.A.G., M.H.-G., E.H.-A. and R.M.H.-A.; visualization, C.A.-G., M.A.G., M.H.-G., E.H.-A. and R.M.H.-A.; supervision, C.A.-G. and M.H.-G. All authors have read and agreed to the published version of the manuscript.

**Funding:** This research received no external funding.

**Institutional Review Board Statement:** This study was conducted in accordance with the Declaration of Helsinki and approved by the Institutional Ethics Committee of the Institute of Neuroscience, University of Guadalajara (protocol code ETl 12015-197, 27 April 2016).

**Informed Consent Statement:** Informed consent was obtained from all subjects involved in the study.

**Data Availability Statement:** The data presented in this study are available on request from the corresponding author.

**Conflicts of Interest:** The authors declare no conflicts of interest.

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
