# Peer review of "Electroencephalografic Activity during the Reading of Erotic Texts with and without Aggression"

_sexes, doi:10.3390/sexes5030016_

Round 1
Reviewer 1 Report
Comments and Suggestions for Authors
Comments
Avoid using “parietal area or temporal area” throughout the text. You should use parietal locations because these are more appropriate for reporting EEG results.
Abstract
- Please describe clearly in the abstract section the results of your research.
Introduction
- I suggest increasing prior papers reporting the brain areas associated with sexual activation.
Methods and Data Analyses
- I recommend conducting a meticulous analysis and computing correlations between the Manikin Self-Assessment Scale and AP of alpha 1 and 2, beta, and gamma.
Discussion
- Please discuss the correlation between the above scale and the AP of each significant frequency.
Reviewer 2 Report
Comments and Suggestions for Authors
Electroencephalografic Activity During the Reading of Erotic or Agresive-Erotic Text
The title needs to be revised. EEG activity seems to be quite vague. Moreover, please, check the English language, since the title is incorrect.
(Electroencephalographic Activity During the Reading of Erotic or Aggressive Erotic Text)
The Abstract contains a vague and old definition of sexual arousal in humans. This needs to be revised.
In the introduction, the authors cited a few brain imaging studies. A more comprehensive review of previous studies must be done and added to the introduction, including the pros and cons of previous studies. Similarly, a recent EEG systematic review from Ziogas (not me) has been published, but it was not included in the text. I believe that could be useful to improve the global quality of the manuscript.
“To date, however, few studies have examined the cerebral basis of SA in women [1, 90 6], and even fewer have used erotic reading with simultaneous EEG recording [16].”- As you can read from different meta-analyses, several studies investigated the SA in women. I agree about a lack of studies that investigated brain underpinnings of cognitive and emotional functions in women, but this needs to be discussed.
The citation of Kinsey is quite old, and I suggest adding new and EEG-related references. Please, remove and rewrite the first paragraphs.
The hypotheses at the end of the introduction need to be stated in a better way.
Methods: The sample size is a limitation. The stimuli used are not sufficiently described, as well as, the general procedure and experimental design. Please, rewrite and add the information as required.
The authors stated that Pearson correlations have been performed, but t values are reported. Similarly, the t values need to be corrected for multiple comparisons. Moreover, please, assess and report the criteria needed to carry out a parametric test.
The discussion is interesting, but I advise you to rewrite it in light of my above- mentioned commentaries (see. Those about the introduction).
Reviewer 3 Report
Comments and Suggestions for Authors
Dear Authors,
I’ve reviewed your manuscript titled " Electroencephalographic Activity During the Reading of Erotic or Aggressive-Erotic Text." I find your study to be a compelling and innovative contribution to the field of sexual arousal research. The exploration of electroencephalographic (EEG) activity in response to sexually explicit texts, particularly the comparison between texts with and without aggression, provides valuable insights into the neural mechanisms underlying sexual arousal and the impact of aggressive content. Your findings on the differential cortical activation patterns and the intricate interplay between cognitive and emotional processes during the reading of these texts are particularly noteworthy. The use of EEG to dissect these nuances represents a significant methodological advancement, offering a fresh perspective that could pave the way for future studies in this domain. However, while your study presents interesting and valuable data, I believe there are several areas where the manuscript could be further strengthened to enhance its clarity, rigor, and overall impact. These improvements will help in better articulating the contributions of your work and in addressing any potential limitations more comprehensively.
Abstract
· The abstract doesn’t clearly state the hypothesis or research question. Explicitly state the hypothesis or research question, such as "We hypothesized that sexually explicit texts with aggressive content (SETA) will generate different cortical responses compared to sexually explicit texts (SET) without aggressive content.
· Provide the average age and relative S.D of participants, “Twenty-seven women participated (mean age, S.D)
· A description of the statistical analyses carried out is completely missing. insert it.
· Terms like "general activation" and "sexual activation" are not defined, which may cause confusion.
· The abstract doesn’t discuss the broader implications or potential applications of the findings. Include a brief statement about the implications of the findings and their potential applications in relevant fields, such as understanding sexual arousal mechanisms or informing therapeutic approaches.
Introduction
· The literature review is extensive but does not focus on the most relevant studies that directly concern current research. Streamline the literature review to focus more on studies that directly concern the effects of aggressive content in erotic texts and the use of the EEG to measure cortical activity during sexual arousal.
· The introduction to EEG techniques is somewhat technical and may be difficult for readers who are not familiar with EEG. Simplify the description of EEG techniques and clearly explain their relevance to the study.
· the introduction does not sufficiently justify the inclusion of aggressive content in the study. provide more scientific evidence on aggressive content (how aggressive content in erotic materials might influence sexual arousal and cortical activation differently from non-aggressive content).
· the introduction doesn’t explicitly state the hypothesis to be tested. clearly state the hypothesis at the end of the introduction 'we hypothesize that sexually explicit texts with aggressive content (SETA) will result in different patterns of cortical activation than sexually explicit non-aggressive texts (SET) in young women'.
Methods
· the sampling method is unclear. please provide more details on the relevant sampling technique used. was an a priori analysis of the ideal sample size carried out?
· It is not clear to me why you opted to recruit only female subjects. it would have been interesting, also in the light of the literature, to look at gender differences. what is the rationale behind this choice?
· The possible presence of sexual dysfunction should also have been assessed through appropriate testing.
· The description of the questionnaires and scales is not thorough and there is a lack of information on the reliability indices. insert Cronbach’s alpha.
· Explain the use of tests to check the normality of the distribution of variables, having first used a non-parametric and then a t test.
· no possible confounding covariates were used in the analyses, I am thinking of analyses with respect to age or level of education as possible influencing factors. to be added.
Results
· Wilcoxon statistics are missing, they are not even included in the tables.
Round 2
Reviewer 2 Report
Comments and Suggestions for Authors
Correct title: Electroencephalographic Activity During the Reading of Erotic Texts with and without Aggression
I found the manuscript improved, but I suggest updating the literature. The authors cited Jakobovits (1965), but as highlighted by van’t Hof and Cera, 2021, Borg, 2014 and other studies, women don’t like hardcore sexual stimuli. This needs to be discussed.
Please, the English language needs to be revised at a professional level.
AS is SA?
“ About, Ziogas et al. [44] carried out a meta-analysis in 146 which they reviewed 255 reported studies covering 81 years of research (1936–2017). This 147 study offers a comprehensive review of global literature on neuroelectric measurements 148 or stimulations associated with various forms of human sexual behavior.” Please, avoid the description and highlight the results. Readers are not interest to effort performed by Ziogas, but they are interested to the results.
Per- 508 haps because, unlike men, women tend to find greater “humanization” in stories or nov- 509 els. The characters have a name, a story and their own desires- this sentence needs to be reformulated.
Author Response
"Please see the attachment."

Reviewer 3 Report
Comments and Suggestions for Authors
Dear authors,
I have had the opportunity to analyse the relevance of what was requested and the changes/additions made, and I can say that valuable and comprehensive revision work was carried out. I therefore believe that, as things stand, this interesting manuscript can be published.
Author Response
Dear reviewer. Thank you for taking the time to review our manuscript. Your suggestions were very helpful to improve the global quality of te manuscript.Round 3
Reviewer 2 Report
Comments and Suggestions for Authors
The authors addressed all the issues that I have raised. However, I suggest changing "brain functionality" to"EEG results" or "neuroelectric correlates" , since "brain functionality" seems to be more related to pathological conditions.
Author Response
Dear reviewer. Thank you for all your suggestions and comments. Without a doubt they were a great contribution to improving the quality of the manuscript.
The recommendation was addressed. You can find it in the discussion on page 18, line 457
